# Sliding of HIV-1 reverse transcriptase over DNA creates a transient P pocket – targeting P-pocket by fragment screening

Abhimanyu K. Singh [1,3], Sergio E. Martinez[1,3], Weijie Gu [1,2], Hoai Nguyen[2], Dominique Schols[1], Piet Herdewijn[2], Steven De Jonghe [1] & Kalyan Das [1✉]

HIV-1 reverse transcriptase (RT) slides over an RNA/DNA or dsDNA substrate while copying the viral RNA to a proviral DNA. We report a crystal structure of RT/dsDNA complex in which RT overstepped the primer 3′-end of a dsDNA substrate and created a transient P-pocket at the priming site. We performed a high-throughput screening of 300 drug-like fragments by X-ray crystallography that identifies two leads that bind the P-pocket, which is composed of structural elements from polymerase active site, primer grip, and template-primer that are resilient to drug-resistance mutations. Analogs of a fragment were synthesized, two of which show noticeable RT inhibition. An engineered RT/DNA aptamer complex could trap the transient P-pocket in solution, and structures of the RT/DNA complex were determined in the presence of an inhibitory fragment. A synthesized analog bound at P-pocket is further analyzed by single-particle cryo-EM. Identification of the P-pocket within HIV RT and the developed structure-based platform provide an opportunity for the design new types of polymerase inhibitors.

[1] Department of Microbiology, Immunology and Transplantation, Laboratory of Virology and Chemotherapy, Rega Institute for Medical Research, KU Leuven, Herestraat 49, 3000 Leuven, Belgium. [2] Department of Pharmaceutical and Pharmacological Sciences, Laboratory of Medicinal Chemistry, Rega Institute for Medical Research, KU Leuven, Herestraat 49, 3000 Leuven, Belgium. [3]These authors contributed equally: Abhimanyu K. Singh, Sergio E. Martinez. ✉email: kalyan.das@kuleuven.be

Antiretroviral therapy (ART) that is used for treating HIV infection generally combines three or more antiviral drugs. Individual antiviral drugs (https://hivinfo.nih.gov/understanding-hiv/fact-sheets/fda-approved-hiv-medicines) have been developed to block different steps of the viral lifecycle—namely the virus attachment and fusion, reverse transcription, integration of viral DNA to the host-cell DNA, and maturation of released viral particles[1]. HIV treatment is life-long and various drug combinations are used to prevent the emergence of drug-resistant viruses such that the plasma viral load in infected individuals remains undetectable. Periodically new drugs are introduced to overcome drug resistance and toxicity that emerge due to the long-term use of existing drugs[2]. Ideally, a new drug is expected to have a mechanism of action that is different from those of the existing drugs. For example, fostemsavir, a gp120 inhibitor, was FDA-approved in 2020 primarily for treating multi-drug resistant HIV infection[3]. Also, new drugs with improved profile against existing targets are being explored; cabotegravir, which is an integrase strand-transfer inhibitor, was approved as the most recent HIV drug. The lessons learned from targeting and suppressing drug resistance in HIV have broader implications in understanding and treating infectious diseases and cancer[4].

HIV-1 reverse transcriptase (RT) copies the viral ~10-kilobase ssRNA genome to a dsDNA in a multistep process[5]. For the DNA synthesis, RT binds an eighteen-base-pair stretch of a duplex substrate with the primer 3′-end nucleotide occupying the priming site (P site). A dNTP complementing the first template overhang binds the N site and the nucleotide part is catalytically incorporated by RT at the 3′-end of the primer with the release of pyrophosphate. Following each nucleotide incorporation, the DNA is translocated from N to P site to accommodate the next dNTP.

Approved RT-inhibiting drugs are non-nucleoside RT inhibitors (NNRTIs) and nucleoside/nucleotide RT inhibitors (NRTIs). NNRTIs allosterically block DNA polymerization by binding a pocket adjacent to the polymerase active site. In general, NRTIs are DNA-chain terminators, i.e., an NRTI being a modified nucleotide is incorporated into a growing DNA strand and blocks the addition of the next nucleotide. RT is responsible for the catalytic incorporation of NRTIs, and RT mutations confer NRTI resistance. The nucleotide analogs are also used for treating infections by RNA viruses that carry RNA-dependent RNA polymerases for transcription and replication of viral RNAs, for example, sofosbuvir[6] and remdesivir[7] are nucleotide analogs used to treat hepatitis C and SARS-CoV-2 infections, respectively.

A polymerase can be directly inhibited by small molecules that would block NTP/dNTP binding to the N site. For HIV-1 RT, the nucleoside-competing RT inhibitors (NcRTIs) have been investigated as potential drug candidates. Unlike an NRTI, NcRTIs do not require cellular phosphorylation steps. NcRTIs compete with dNTPs for binding at the N site and can be categorized as metal-dependent inhibitors such as α-CNP[8–10] and metal-independent inhibitors such as INDOPY-1[11,12]. Despite these developments, the discovery of new druggable sites of RT that are highly conserved is important for developing new classes of drugs with resilience to existing drug-resistance mutations.

In general, the transient states have been considered as important targets for drug design[13], and there may exist unidentified transient states of HIV-1 RT that can be valuable for discovering new classes of drugs. During the polymerization process, RT has been shown to slide over its substrate by single-molecule FRET studies[14,15], which indicates potential for the existence of transient states, however, no transient state has been structurally characterized. In this study, we crystallized HIV-1 RT/dsDNA cross-linked complex[16] in a new crystal form with two copies of the complex present in the crystallographic asymmetric unit: in one, the primer 3′-end nucleotide occupies the N site representing the pretranslocation complex (N complex) and in the other copy, RT slides ahead of dsDNA such that the primer 3′ terminus occupies P-1 site (P-1 complex). A transient pocket is formed at the P site in this P-1 complex. We refer the pocket as P-pocket hereafter. Fragment screening is a valuable experimental technique to find new druggable pockets or new chemical scaffolds for binding to an existing pocket[17]. We screened 300 fragments from the DSi-Poised[18] and Fraglites[19] libraries by X-ray crystallography using the state-of-the-art facility Xchem[20] at Diamond Light Source (UK) and identified two compounds (048 and 166) binding at P-pocket, which is flanked by highly conserved structural motifs. We performed structure-based virtual screening of related compounds and synthesized five close analogs of fragment 166. Using a modified DNA aptamer, we trapped the P-1 state in solution with P-pocket available for inhibitor binding. Structures of the hit fragment 166 and a newly synthesized analog F04 bound at P-pocket were determined by single-particle cryo-electron microscopy (cryo-EM).

Our study revealed a highly conserved transient P-pocket that is created in the process of sliding of RT over a dsDNA substrate and established an experimental platform for structure-based drug design using single-particle cryo-EM. This finding opens the possibility for discovering new classes of RT inhibitors, and may help extend the concept for identifying and targeting analogous pockets in other viral polymerases.

## Results

**N- and P-1 complexes coexist in the crystal**. In the current study, we crystallized the binary complex of I63C mutant RT cross-linked to a 28/21-mer dsDNA template/primer (Fig. 1a) in a new crystal form (Supplementary Table 1) that contains two copies of the complex, and the structure was determined at 2.85-Å resolution. Earlier, the I63C RT cross-linked with a 27/21-mer dsDNA was crystallized as a polymerase active complex in which, the primer 3′-end occupied the P site (P complex; Fig. 1b) and an incoming d4T-TP was bound at the N site,[16] the extra nucleotide in the 28/21-mer DNA is on the template overhang, which is primarily disordered in the structures. No dNTP was added to our current crystallization and therefore, we expected that both copies in the crystal were of P-complex. Surprisingly, we observed that in the first copy, RT has backtracked over the DNA by one-nucleotide length such that the primer 3′-end nucleotide occupies the N site, and the structure represents the state following a nucleotide incorporation and prior to translocation (N complex, Fig. 1c). Functionally, RT backtracks as an N complex for excision, the reverse reaction of polymerization that RT facilitates to excise nucleoside analogs such as AZT from the primer 3′-end[21–23]. In the second copy in the crystallographic asymmetric unit, RT has slipped ahead of the DNA substrate by about one-nucleotide length such that the primer 3′-end occupies P-1 site (P-1 complex, Fig. 1d).

**I63C cross-link permits sliding of RT over a dsDNA substrate**. For structural studies of catalytic-competent RT/dsDNA complex, RT is usually cross-linked to a modified DNA base. The predominantly used cross-linking is done between a mutated residue Q258C of the thumb subdomain and a modified guanine base of template or primer, which is the fifth nucleotide from the P site[24,25]. The Q258C cross-linking prevents sliding of RT over DNA and arrests a stable complex for structural studies. The Q258C cross-linking was used to obtain the structures of both N- and P-complexes of RT/DNA when the appropriate DNA substrate was cross-linked[22]. Similarly, the N- and P-complexes have

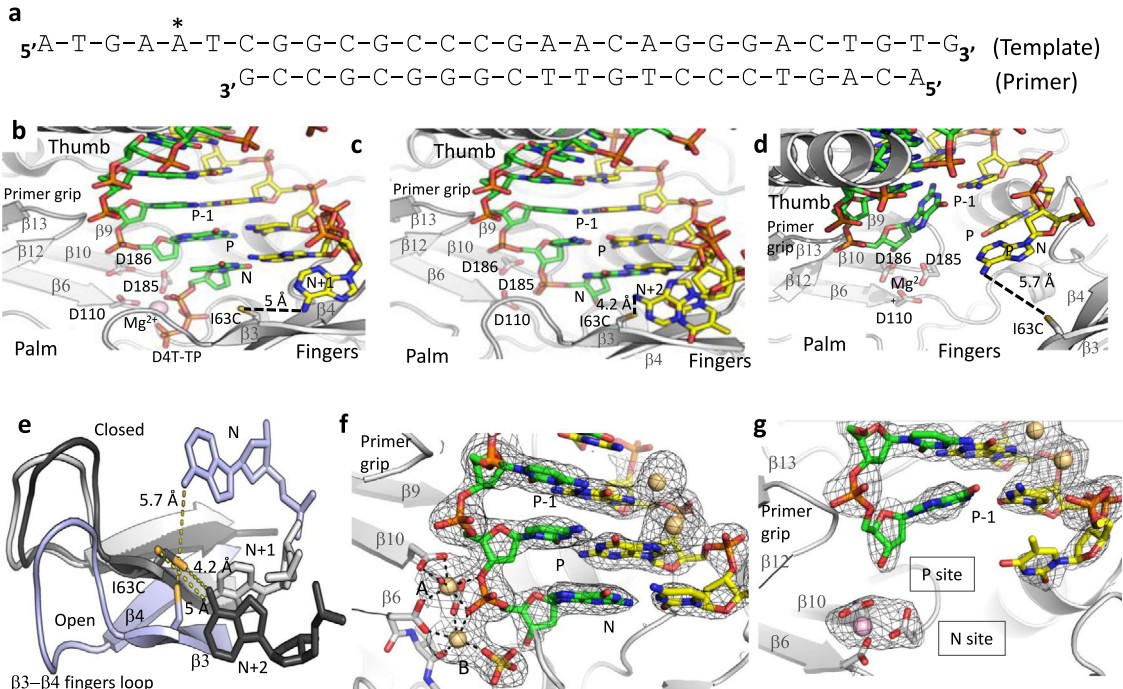

**Fig. 1 Sliding of RT and trapped P-1 and N complexes in crystals. a** The 28/21-mer template primer used in the I63C RT/dsDNA cross-linked complex, the cross-linked site in the template is marked (*). **b** The catalytic complex (P complex) of RT/DNA in which the primer 3′-end nucleotide occupies the P site and a dNTP or analog binds the N site. The shown structure (PDB ID 6AMO)[16] has a d4T-TP bound at the N site and the 3′-dideoxy-GMP-terminated primer prevents the catalytic incorporation of d4T; the cross-link between I63C and the modified dAMP at the template N + 1 position is symbolically indicated by a dotted line. The DNA template primer is in yellow and green, respectively. The residues D110, D185, and D186 define the catalytic triad. **c** The polymerase active site of the N complex as observed in the first copy in the crystal. The RT slides back such that the primer 3′-end nucleotide is at the N site and the cross-linked dAMP is now at the template N + 2 position. **d** Second copy in the crystal represents the P-1 complex in which the primer 3′-end has moved away from the active site by almost one-nucleotide distance. The P-1 complex is formed by RT sliding ahead of DNA; the cross-linked dAMP is now occupying the template N position. **e** The relative positions of the cross-linked dAMP and I63C in N (dark gray), P (light gray), and P-1 (light blue) complexes, three structures were aligned based on the superposition of the palm subdomains. Because of the unoccupied dNTP-binding site in the P-1 complex, the fingers subdomain is in an open conformation. **f** 2Fo–Fc electron density map defines the track of the DNA near the polymerase active site in the N complex. The Cd²⁺ ions are shown as light-brown spheres. Two Cd²⁺ ions chelate the catalytic aspartates, DNA primer, and a sulfate ion at the active site; however, seven coordination of ion A and five coordination of ion B suggest that Cd²⁺ ions at the active site do not facilitate catalysis. **g** 2Fo–Fc electron-density map defines the track of the DNA in the P-1 complex; a Mg²⁺ ion at the active site is shown as pink sphere.

been also trapped using a DNA aptamer that resists RT translocation[9,26]. In contrast, the structures of N, P, and P-1 complexes (Fig. 1b, c, d) achieved by using I63C RT/DNA cross-linked complex reveal that the flexible fingers subdomain and the single-strand part of the DNA on both sides of the cross-linked disulfide bond permit sliding of RT over the dsDNA substrate by at least two-nucleotide lengths. The cross-linked dAMP on the template overhang occupies N + 2, N + 1, and N positions in the structures of N, P, and P-1 complexes, respectively (Fig. 1e). The N6–Sγ (Cys63) linker distances in P, N, and P-1 complexes are 5, 4.2, and 5.7 Å, respectively; the distance in P-1 complex is the longest among three complexes yet, 5.7 Å can be spanned by a N–C–C–S–S linker.

**Cd²⁺ ions arrest N complex and DNA:DNA crystal contact stabilizes P-1 complex.** The crystals of I63C RT/DNA complex unexpectedly contain an N complex and a P-1 complex in the crystallographic asymmetric unit rather than two copies of the P complex. What is the basis for the structural heterogeneity in the current crystals which is rather uncommon? In general, a protein or complex is crystallized at a low energy state. For DNA synthesis, RT/dsDNA in solution is required to exist as the P complex which permits the binding of an incoming dNTP at the N site. Following the nucleotide incorporation, the N complex is

formed and then translocated to the P complex. These complexes in solution, however, are in dynamic equilibrium with a higher probability of existing as the P complex for accommodating an incoming dNTP. Apart from attaining the N complex, RT/DNA is also expected to visit multiple transient states that are short-lived. The dynamic behavior of RT flipping and sliding over a dsDNA or an RNA/DNA substrate has been observed by single-molecule FRET studies[14,15,27].

Our current crystallization requires Cd²⁺ ions and the structure revealed that two active-site Mg²⁺ ions in the N complex have been replaced by two Cd²⁺ ions (Fig. 1f); significantly strong scatterer Cd²⁺ ions that have 46 electrons compared with Mg²⁺ ions that have 10 electrons produced strong electron density peaks, and Cd²⁺ ions were unambiguously located in the structure. The structure implies that higher chelation affinity of Cd²⁺ ions compared with that of Mg²⁺ ions stabilized the RT/DNA as the N complex in the first copy. Biochemically, Cd²⁺ ions have been shown to inhibit DNA polymerization by RT even at a low concentration of 1–10 μg/ml[28]. In the crystal structure, two active-site Cd²⁺ ions trap the N complex and block translocation. Also, the chelation geometry of both Cd²⁺ ions at the active site deviates from the classic octahedral chelation geometry observed for Mg²⁺ ions in a catalytic active complex[29] suggesting that the Cd²⁺ ion chelation is not competent for carrying out catalytic reaction. Strong yet

nonproductive chelation of $Cd^{2+}$ ions at the active site appears to be responsible for RT inhibition.

In the current crystal structure, we observed that the other end of DNA of this N-complex structure has extended beyond the RNase H active site and interacts with the DNA of the second copy of the complex in the asymmetric unit (Supplementary Fig. 1). The 3′-end of the template has a dGMP overhang beyond the DNA-duplex region, and the guanine base of this nucleotide in the first copy is intercalated between the duplex DNA and the 3′-end dGMP of the second copy and vice versa for the 3′-end guanine overhang of the second copy. This DNA:DNA interaction at the non-crystallography-symmetry interface, which is not present in the previously reported P-complex structure of the I63C RT/DNA[16], appears to be primarily responsible for stabilizing the P-1 complex of RT in the second copy while the first copy is stabilized as the N complex by the active site $Cd^{2+}$ ions (Fig. 1f, g). An experiment to replace the $Cd^{2+}$ with $Mg^{2+}$ ions by transferring the crystals to the crystallization buffer without $CdCl_2$ dissolved the crystals immediately suggesting that $Cd^{2+}$ ion is essential for the stability of the crystal. Also, crystals never grew in the absence of cadmium.

**Fragment screening discovers small-molecule binding to P-pocket.** The sliding of RT/DNA to the P-1 complex (Supplementary Movies 1 and 2) creates a transient P-pocket, and the crystal-lattice contacts stabilize the open-pocket conformation, suggesting that the energy required for creating and stabilizing this pocket is not high. P-pocket is located between two key structural elements—the active site YMDD motif and the primer grip, both are conserved and essential for the DNA polymerization by HIV-1 RT. Blocking this pocket by small molecules would inhibit RT. We used a one-of-a-kind high-throughput XChem fragment screening facility at the Diamond Light Source, UK, to find chemical scaffolds that can bind P-pocket (Fig. 2). We used a subset of 300 fragments from FragLites[19] and brominated DSI-Poised[18] libraries provided by the XChem facility. The subset was chosen for its high solubility in ethylene glycol as our crystals were more resilient to ethylene glycol than to organic solvents like DMSO. The diffraction datasets were collected unattended using the automated data collection setup at the I04-1 beamline. The datasets were processed through the automated data-processing pipeline available at I04-1. However, owing to their moderate resolution, only thirty best datasets were selected for manual processing, structure solution, and difference map calculations. The analysis of the diffraction datasets revealed the binding of two fragments **166** and **048** to P-pocket (Fig. 2). The detailed experimental protocol is described in the experimental section and schematically shown in Supplementary Fig. 2.

**Binding of fragment 166.** Fragment **166**, ((1 R,2 R)-2-phenyl-N-(1,3-thiazol-2-yl)) cyclopropanecarboxamide (Fig. 3a), occupies P-pocket. Electron density for this fragment is clear and consistent with the chirality of the compound (Fig. 3b). The interacting amino acid residues and nucleotides are shown in Fig. 3c, and a 2D representation of the interactions of **166** is shown in Fig. 3d. The binding of **166** induced reorganization of P-pocket (Supplementary Movie 3). A comparison of the **166**-bound structure with the RT/DNA P-1 complex structure with no compound bound, hereafter referred as the apo structure, shows the expansion of the pocket upon fragment binding (Fig. 3e). The deoxyribose ring of the primer 3′-end nucleotide has shifted by ~2.7 Å to accommodate **166**. The five-membered thiazole moiety of **166** interacts with the deoxyribose ring and with the primer grip. The central cyclopropyl moiety is positioned over the conserved YMDD motif of the polymerase active site and interacting

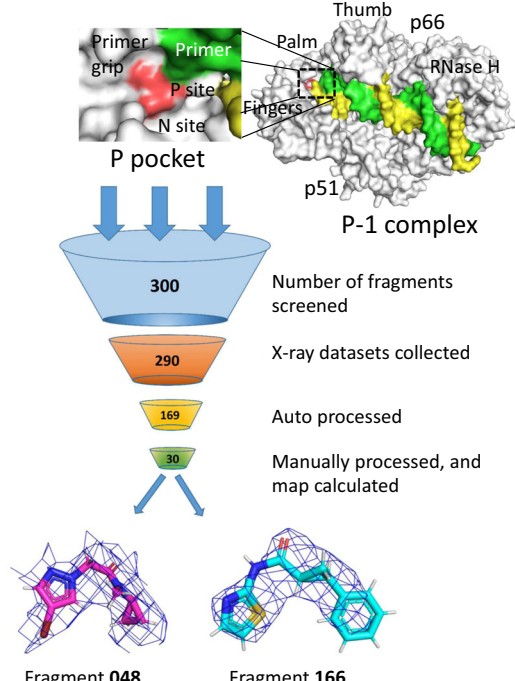

**Fig. 2 A schematic overview of the fragment-screening protocol.** The RT/DNA P-1 complex structure in the right shows the location of P-pocket, which is zoomed in the top-left panel. Steps of data collection and processing starting from the preparation of crystals of three hundred fragment complexes with HIV-1 RT/DNA. Thirty diffraction datasets with useful resolution were obtained of which electron density was observed for binding of two fragments, **048** and **166**; the Fo–Fc difference maps covering the fragments are displayed at 2.5σ.

with the active-site residues Y183, M184, and D185. Consequently, the YMDD motif has bent down by ~1.6 Å compared with the apo structure. One $Mg^{2+}$ ion that was chelating all three catalytic aspartates (D110, D185, and D186) in the apo structure, is dissociated upon binding of **166**. The primer grip has moved away from the inhibitor by ~1 Å, which is the least among all structural motifs forming walls of P-pocket. The phenyl ring of **166** points toward the first template-overhang thymine base. This thymine base, which is less ordered in the apo structure due to lack of interaction, has shifted by ~2.2 Å and aligned with the phenyl ring of **166**. Despite no canonical base-pairing or a typical hydrogen-bond interaction between the thymine base and the phenyl ring of **166**, both aromatic rings are aligned and stacked against the dsDNA base pair (Fig. 3e), suggesting that modification of the phenyl ring of **166** to form base-pair-like interaction with the template base would improve binding. Compound **166** in P-pocket almost mimics the binding of a nucleotide (Fig. 3f) and the predominant interaction of **166** is its stacking with the terminus base pair at the P-1 position.

**Binding of fragment 048.** Like fragment **166**, the fragment **048** (2-(4-bromo-1H-pyrazol-1-yl)-N-cyclopropyl-N-methylacetamide; Fig. 4a) binds P-pocket with a clear electron density (Fig. 4b). Fragment **048** is smaller in size and contains two rings —a cyclopropyl and a pyrazole. The cyclopropyl group is the only chemical moiety common to both fragments **048** and **166**, and the cyclopropyl group occupies a common area of P-pocket in both structures, despite that two fragments fill different parts of the pocket. The cyclopropyl ring is positioned over the YMDD motif in the pocket (Fig. 4c), and like **166**, fragment **048** binding knocks out the active site $Mg^{2+}$ ion. The 4-bromopyrazole ring

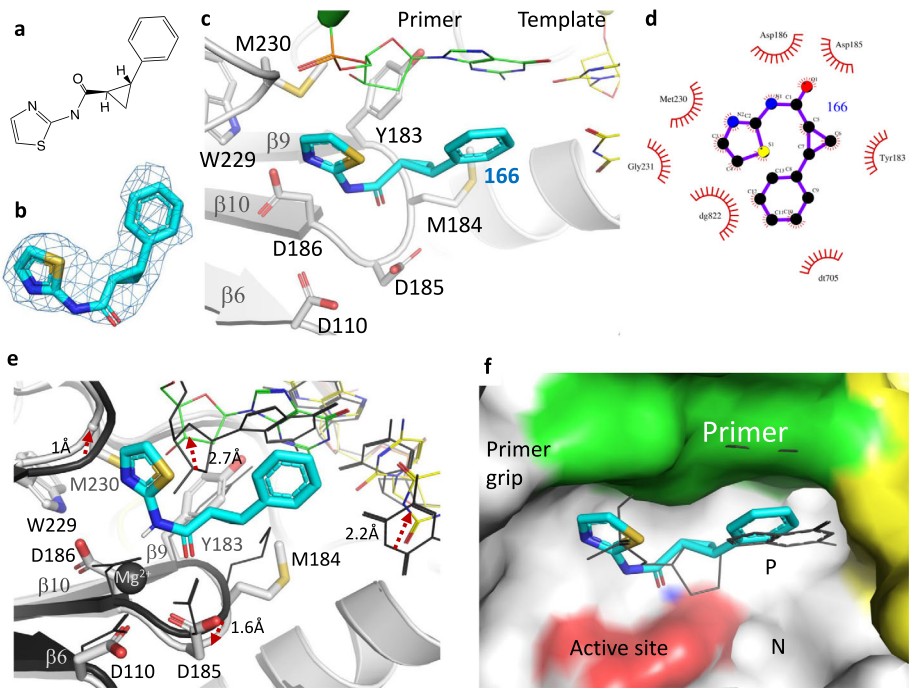

**Fig. 3 Binding of fragment 166 to P-pocket. a** Chemical structure of **166. b** Polder map contoured at 6σ defines the fitting of **166** in P-pocket. **c** The fragment **166** (cyan) in P-pocket of HIV-1 RT (gray ribbon); the nucleotides and protein residues surrounding **166** are shown. **d** LigPlot showing a 2D view of the interactions of **166** with nearby residues/nucleotides. **e** The binding of **166** caused expansion and rearrangement (Supplementary Movie 3) as revealed by comparing P-pockets of apo P-1 complex (dark gray) and **166**-bound P-1 complex (light-gray RT, green primer, and yellow template); the shifts of key motifs in the pocket are indicated by dotted red arrows. The rmsd for the pocket residues is 1.4 Å. The $Mg^{2+}$ ion in the apo structure is dissociated upon the fragment binding. **f** A superposition of structures of RT/DNA/d4T-TP P complex (PDB ID. 6AMO) and RT/DNA/**166** P-1 complex shows that **166** mimics the positioning of a nucleotide in P-pocket. The shown molecular surface is of RT/DNA/**166** P-1 complex.

interacts with the residues Y115 and Q151 (Fig. 4d) and points toward the dNTP-binding site (or N site). The 4-bromopyrazole moiety is stacked against Y115 sidechain and the bromine atom interacts with the Q151 sidechain. The pocket rearrangement is less extensive for accommodating **048** (Fig. 4e) compared with **166**, which indicates elasticity of this transient P-pocket. The deoxyribose ring of the primer 3′-end nucleotide and the YMDD motif each shifts by ~1.3 Å for binding **048** when compared with the apo structure. A superposition of **166**- and **048**-bound RT/DNA structures suggests the potential for P-pocket to accommodating different chemical moieties at different parts (Fig. 4f). The B-factors of the fragments and surrounding residues are highly comparable; the average B-factor of the fragment **048** is 79.60 Å$^2$ while the P-pocket residues have an average B-factor of 75.67 Å$^2$, and the average B-factor of fragment **166** is 122.30 Å$^2$, while the pocket has an average B-factor of 108.39 Å$^2$. These data suggest that the P-pocket is flexible yet, nearly fully occupied by the fragments.

**Fragment-based design—in silico docking and synthesis**. To investigate if the newly discovered P-pocket can accommodate analogs of **166** with potential for improving binding affinity, we initiated a docking study targeting P-pocket in the RT/DNA/**166** structure. The aim was to evaluate various chemical substitutions to the azole–cyclopropyl–phenol backbone that mimics the binding of a nucleotide (Fig. 3f). We started our post-fragment soaking study by optimizing fragment **166** because (i) it occupies a larger part of the pocket, (ii) its binding caused a larger structural rearrangement of the P-pocket than **048**, and (iii) **166** interacts with highly conserved structural elements, YMDD motif and primer grip. The rationale behind the design of a virtual library of analogs of 166, is described in Supplementary Methods

section "Fragment design and docking study". A total of 84 fragments were designed virtually with the aim to (i) form base-pair-like interaction with the template first overhang dTMP, (ii) engage one or more invariant catalytic aspartates or (iii) both (Supplementary Table 2). Fragments **F01–F18** were designed by substituting the phenyl ring of **166** with a pyridine moiety and introducing an amino or hydroxyl arm at the 6′-position of the pyridine ring. We maintained the amide bond of **166** and replaced the thiazole moiety with different five-membered heteroaromatics. Docking of **F01–F18** to P-pocket of the RT/DNA/**166** complex showed favorable pseudo-base-pairing interactions for the pyridine ring. Among those, **F01–F05** were of particular interest owing to their synthesis convenience. The docking results suggested better docking scores for **F01–F05** than the parent **166** (Supplementary Table 3 and Supplementary Fig. 3). Detailed synthesis for **F01–F05** is outlined in Supplementary Methods section "Synthesis of fragments".

Next, in an attempt for engaging the catalytic aspartates, fragments **F19–F84** were designed where the amide bond was substituted for an amidine or guanidine linker (Supplementary Table 2). The docking result suggested that amidine and guanidine can form hydrogen bond or salt-bridge interactions with the catalytic residues D185 and/or D186 and two fragments, **F47** and **F81**, were selected for synthesis (Supplementary Fig. 4). In addition, newly introduced heteroaromatics have the potential to form additional hydrogen bonds with surrounding residues. Synthesis of **F47** or **F81** has not been successful yet.

**Cryo-EM structures of RT/DNA aptamer in complexes with 166 and F04.** In our fragment screening-experiment, we could get only thirty X-ray diffraction datasets from 300 fragment-soaked crystals (Fig. 2). In subsequent studies, we experienced that often

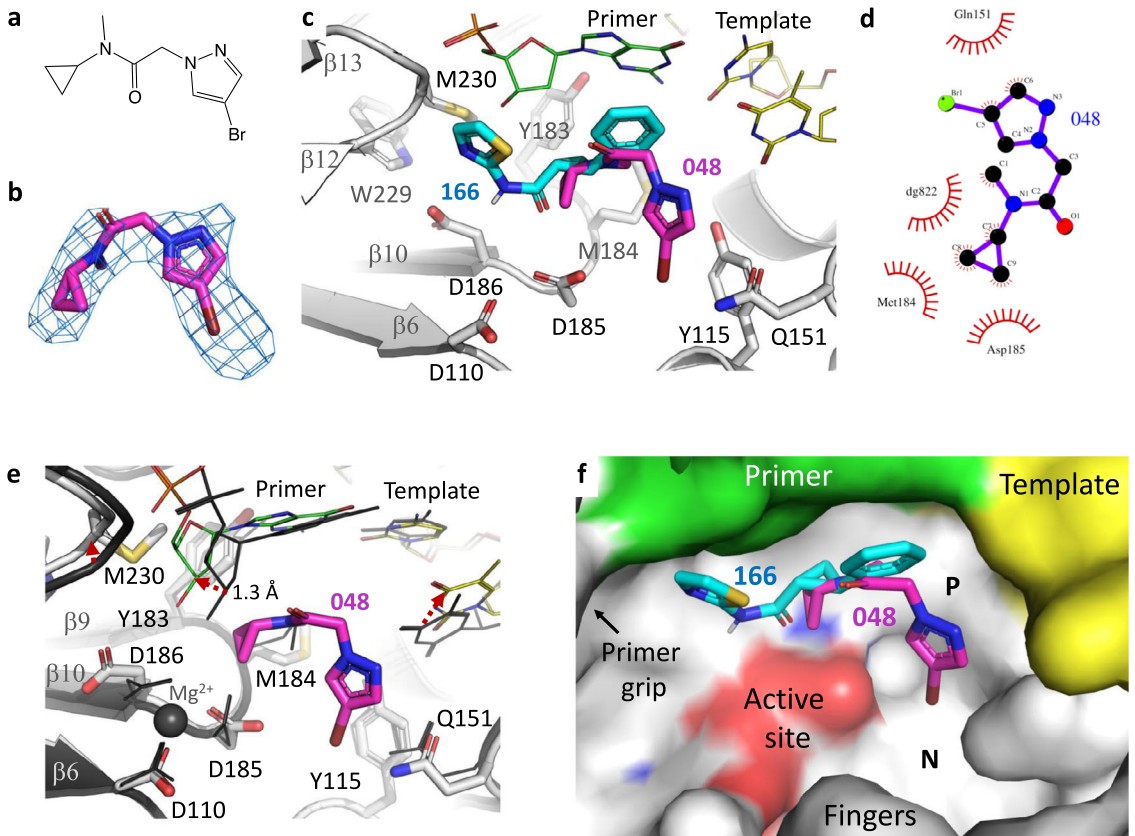

**Fig. 4 Binding of fragment 048 to P pocket. a** The chemical structure of fragment **048**. **b** Polder map at 4.5σ contour defines the binding mode of **048** to P-pocket. **c** The binding mode of **048** is distinct from that of **166**. **d** LigPlot showing a 2D representation of interactions of **048** with nearby residues and nucleotides. **e** The binding of **048** caused expansion and rearrangement as revealed by comparing P-pocket of apo P-1 complex (dark gray) and **048**-bound P-1 complex (light-gray RT, green primer, and yellow template); the shifts of key motifs in the pocket are indicated by dotted red arrows. The rmsd for the pocket residues is 1.2 Å. **f** Fragments **166** and **048** occupy different parts of P-pocket.

the crystals are either dissolved or losing diffraction quality in soaking experiments that hinder our ability to routinely and reliably obtain the structures for this drug-design project. It is likely that soaking of the compounds interferes with the $Cd^{2+}$ ion interactions and destabilizes the crystals. The $Cd^{2+}$ ions play a significant role in stabilizing the N- and P-1 complexes in the crystal. Twenty-seven ordered $Cd^{2+}$ ions are located in the asymmetric unit, and $Cd^{2+}$ ions interact with DNA template and primer near P-pocket (Fig. 1g). To overcome this experimental limitation, we investigated the possibility of using single-particle cryo-EM to the project. The cryo-EM structures are now free from the impact of $Cd^{2+}$ and crystal contacts.

Single-particle cryo-electron microscopy (cryo-EM) has emerged as an effective technique for structural studies of macromolecular systems at or near atomic resolution[30], and the technique can be powerful for structure-based drug design[31] when a sample is optimized for generating good-quality vitreous grids reproducibly, by efficient data acquisition, and fast processing. In addition, for current target, we required to trap the RT/DNA P-1 complex in solution for cryo-EM studies. The following paragraph explains the rationale for trapping the complex carrying P-pocket in solution state.

A selective evolution of ligands by exponential enrichment (SELEX) based screening discovered a nucleic acid template:primer aptamer that binds RT with significantly higher affinity than a typical template:primer substrate[32]. A 38-mer DNA aptamer that binds RT at subnanomolar affinity folds as a 15-mer duplex that contains two 2′-O-methylated nucleotides at −2 and −4 template positions, and contains a hairpin of three thymine

nucleotides (3-T) at positions 16–18[33]. The reported crystal structures of the 38-mer DNA aptamer in complex with RT revealed that the interactions of the 2′-O-methyl groups and the 3-T hairpin contribute to the higher stability of the complex that also prevents sliding of RT over the DNA aptamer[9,26]. Based on this finding, we used a 37-mer DNA aptamer that lacks the first 3′-end nucleotide (Supplementary Fig. 5a) for trapping the RT/DNA P-1 complex. The complex was purified over a size-exclusion column and showed homogeneous particle size distribution by dynamic light scattering ("Methods" section; Supplementary Fig. 5). The cryo-EM structures of this complex were determined with bound fragments **166** and **F04** at 3.38- and 3.58-Å resolution (Supplementary Table 4), respectively. High-resolution cryo-EM structures of RT/dsRNA representing a minimum transcription-initiation complex have been reported recently[34]. The structures were achieved by acquiring and processing a significantly large quantity of data. In contrast, our cryo-EM condition optimization was focused on fast data collection and processing, which is essential for a structure-based drug-design project.

The overall conformation of RT and the track of DNA in RT/DNA aptamer/**166** P-1 complex cryo-EM structure align well with the crystal structure of RT/DNA-aptamer complex[26] (Supplementary Fig. 6a), suggesting a minimum impact of the experimental conditions of earlier crystallization or current cryo-EM grid preparation on the protein and DNA aptamer. A primary aim for determining the structure of RT/DNA aptamer/**166** P-1 complex by both cryo-EM and crystallography is to assess the impacts of two distinct techniques on (i) the

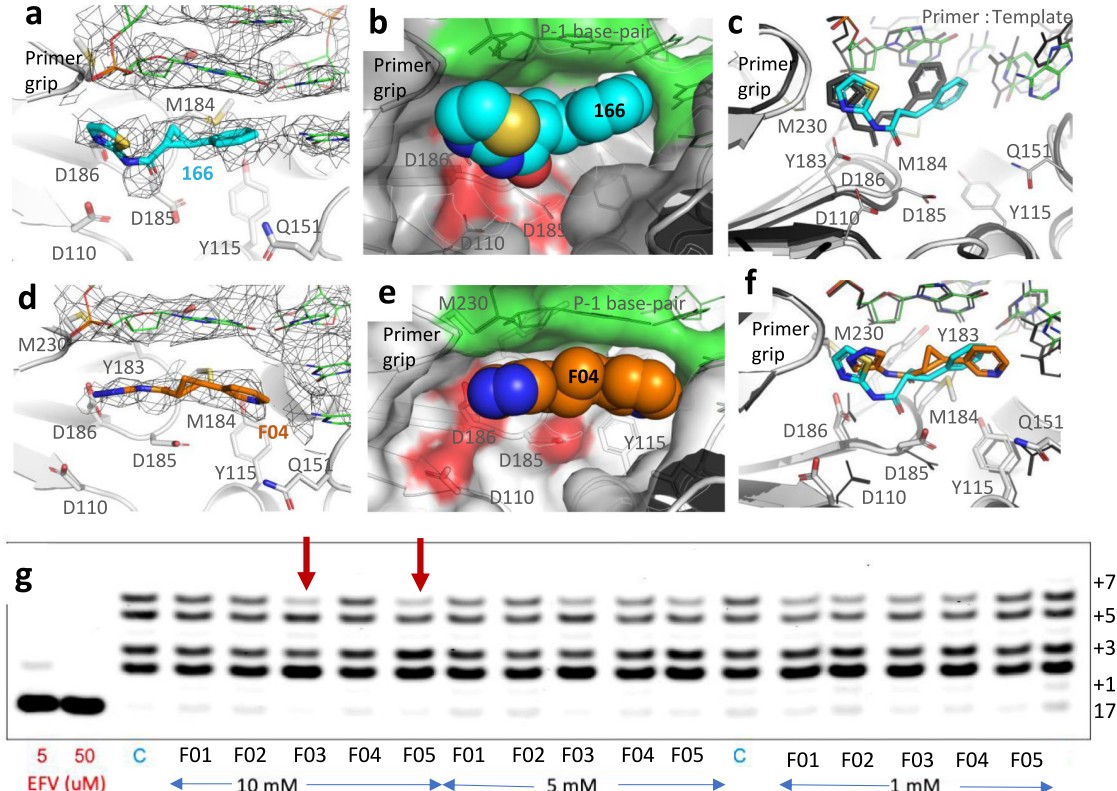

**Fig. 5 Cryo-EM structures show binding of fragments at P-pocket and RT inhibition by fragments. a** The 3.38 Å resolution single-particle cryo-EM density map reveals the binding of **166** at P-pocket; the density map for **166** in stereo is shown in Supplementary Fig. 9. **b** The space-filling model of **166** in P-pocket shows that the fragment stacks with the DNA (green surface). The RT part of the pocket is in gray with the active site catalytic residues spotted red. **c** Comparison of the mode of binding of **166** to P-pocket in the crystal structure (dark gray) and cryo-EM structure (cyan fragment, gray RT, and green DNA) reveal a similar mode of binding, however, nonsuperimposable pockets may be attributed to two distinct structure determination techniques; the presence of $Cd^{2+}$ ions in the crystal (Fig. 1g) might be partly responsible for the shifts in the pocket. **d** The 3.58 Å resolution cryo-EM density map shows the binding of synthesized fragment **F04** to P-pocket; the density map for **F04** in stereo is shown in Supplementary Fig. 9. **e** The space-filling model of **F04** shows the positioning of the fragment in P-pocket including the stacking interaction with DNA. **f** A comparison of the binding modes of **F04** (orange **F04**, light gray RT, and green DNA) with **166** (cyan **166**, dark gray RT and DNA) based on superposition of two cryo-EM structures reveals differences in the modes of binding of two fragments to a highly superimposable P-pocket. **g** HIV-1 RT inhibition assay carried out using a Cy5 fluorophore-labeled primer and template with seven dT overhangs. The reaction mixture had 125 nM primer-template, 50 mM Tris-HCl, pH 8.3, 3 mM $MgCl_2$, 10 mM DTT, and 5 μM dATP. The inhibition data are shown for efavirenz (positive control inhibitor), C (no inhibitor), and fragments **F01–F05** (at 10, 5, and 1 mM) concentration; the red arrows indicate that the fragments **F03** and **F05** show RT inhibition at 10 mM concentration. As labeled in the right, the gel has a single-nucleotide incorporation resolution starting from 17 to 24 nucleotides. Supplementary Fig 10A shows a ladder with 15, 17, 23, and 30 nucleotide markers for calibration. The raw gel is shown as Supplementary Fig 10b. The experiments were performed reproducibly in triplicate.

characteristics of the transient pocket and (ii) binding of the fragment that has a relatively weak affinity. The cryo-EM density map revealed the binding of fragment **166** to P-pocket (Fig. 5a) where **166** stacks with the P-1 DNA base pair as in the crystal structure (Fig. 5b). A superimposition of the structures of RT/ DNA aptamer/**166** P-1 complex determined by cryo-EM and crystallography in the current study (Supplementary Fig. 6b) shows that the conformation and the mode of binding of **166** in two structures are similar, however, not identical (Fig. 5c). The observed structural changes may be attributed to the distinctly different experimental conditions. Considering that the P-1 complex represents a transient state, involvement of $Cd^{2+}$ ions with DNA nucleotides (Fig. 1g) and the crystal symmetry interfaces apparently has some influence on the conformation of the transient P-pocket. However, the overall protein fold, domain structures, and track of DNA align well (Supplementary Fig. 6b) between the RT/DNA/166 structure determined by cryo-EM and X-ray crystallography.

The cryo-EM density in P-pocket of RT/DNA aptamer/**F04** complex clearly defined the location and confirmation of the

fragment (Fig. 5d). The fragment **F04** was synthesized as a racemic mixture, and the cryo-EM density suggests that binding of the (R,R)-enantiomer is favored. The fragment was predicted to have hydrogen-bond interaction with thymine base of the first template overhang. In the structure, although the pyridine ring pointed towards the thymine base of dTMP, it failed to develop a strong H-bond with the base. **F04** is stacked with the first base-pair of DNA duplex at the P-1 position (Fig. 5e), and this stacking interaction appears to be improved when compared with the stacking of **166**. Better density for **F04** than for **166** in their respective cryo-EM structures (Supplementary Fig. 7) may relate to the improved stacking interactions of **F04** with the P-1 base pair. Because of almost identical experimental conditions, the RT and DNA conformations in both **166** and **F04** complexes are highly superimposable even at their P-pocket, both structures align on Cα atoms with rmsd of 0.74 Å (Fig. 5f). Thereby, the difference in the positioning of the fragments in the pocket may be attributed to the differences in their chemical structures.

The discussed structures primarily represent states of elongation complex. Structures of HIV-1 RT transcription initiation

complexes that bind a dsRNA substrate show significantly different track for dsRNA and the position of primer 3′-end when compared with RT/dsDNA elongation complex[34–36]. A comparison of P-1 state with RT/dsRNA complexes (Supplementary Fig. 8) shows that the primer P-1 nucleotide of dsRNA is about 5 Å away from the DNA primer 3′-end at P-1 position. Due to this structural difference, a potential P-pocket in RT/dsRNA complex will be different from the P-pocket observed in the current study.

**RT inhibition**. Our current fragments are weak binders that lack significant specific interactions with P-pocket residues. Moreover, for RT polymerase assay, the presence of dNTP would shift the equilibrium of RT/DNA to P complex that would compete with the formation of transient P-pocket and binding of a fragment to the pocket. Addition of dNTP would also increase the rate of dissociation of a bound fragment. The above facts suggest that high-affinity binders of P-pocket are required for reliable measurement of inhibition. For our current fragments, we used a qualitative RT gel-based incorporation assay[37] to examine if any of the synthesized fragments shows some inhibition of DNA polymerization by RT; details are in experimental section. Our assay demonstrated weak, but noticeable, RT inhibition by the fragments **F03** and **F05** at 10 mM concentration advocating P-pocket as a promising druggable pocket (Fig. 5g). Compounds **F01–F05** have a common pyrimidine ring replacing the phenyl ring of **166**, whereas different 5-membered heterocycles are substituted for the thiazole ring. The RT-inhibition assay results show that a pyrazole ring or a 1,2,4-triazole is favorable for RT inhibition. The data in the realm of the structure of **F04** complex suggest that a nitrogen at position 1 can form a hydrogen-bond with D186.

## Discussion

RT was shown to slide over an RNA/DNA or a dsDNA substrate in the process of DNA synthesis. Here for the first time, we trapped a transient state in which RT has slid ahead of DNA by one-nucleotide distance and created a pocket, P-pocket. In the absence of dNTP, the binary I63C RT/DNA cross-linked complex was more flexible in solution, and the crystal contacts helped trap a transient state of RT/DNA complex that carried P-pocket. Using X-ray crystallography, we screened 300 small-molecule fragments and discovered the binding of two fragments to P-pocket. The mode of binding of fragment **166** mimics the positioning of the primer 3′-end nucleotide in the catalytic P complex. The binding of **166** suggested a potential for pseudo-base-pair-like interaction with the first template overhang at one end and expansion of the fragment to gain interaction with the catalytic triad (D110, D185, and D186) from the opposite end. We designed such compounds virtually and docked those in P-pocket. Five selected compounds with simple chemical modifications and improved docking scores compared with **166** were synthesized, and two of those showed inhibition of DNA polymerization by HIV-1 RT. We used an RT/DNA aptamer complex that trapped the transient P-pocket in solution without needing help from crystal lattice contacts or Cd$^{2+}$ ions that were required for crystallization. The RT/DNA aptamer in complexes with two fragments was optimized on cryo-EM grids and the structures of the complexes were determined from single-particle data collected using an in-house Glacios 200 kV transmission electron microscope. The optimization of samples on cryo-EM grids and data collection protocol reproducibly yielded high-quality data from less than one thousand micrographs. These optimizations are essential for high throughput structural study for a drug-design project using single-particle cryo-EM.

The current fragments provide a basic chemical scaffold that can bind the transient P-pocket and thereby the study discovers a previously unknown druggable site of HIV-1 RT. There exist multiple opportunities to design compounds with pocket-specific interactions such as pseudo A:T or G:C base-pairing and extension to reach active site, primer grip, and key residues such as Y115 and M151 that are involved in dNTP binding; binding of fragment **048** indicates the potential for acquiring interactions with Y115 and M151. While our study provides the basis for exploring an innovative site of RT for drug design, the current fragment scaffold that mimics a nucleotide may also have a broader implication for finding inhibitors of RNA polymerase to inhibit by an analogous mechanism. This study also advocates for utilizing combinations of powerful biophysical and biochemical techniques in uncovering and validating transient pockets.

## Methods

**RT expression, RT/DNA cross-linking, purification, and crystallization**. The RT containing I63 C mutation (RT139A) was constructed and purified as described[16]. Briefly, after Ni-NTA purification, the His-6 tag was cleaved from the N terminus of p51 using 1:10 mass ratio of GST-tagged HRV14 3C protease to RT, overnight on ice. Purification continued with Mono Q.

To generate a 28-mer template with a cross-linking dA at the second-base overhang, the oligomer 5′-ATGA**A**TCGGCGCCCGAACAGGGACTGTG-3′ was ordered from Midland Certified Reagent Company in Midland, TX,; the underlined, bold A indicates the O$^6$-phenyl-dI base. The O$^6$ phenyl group was substituted with a cystamine group (H2NCH2CH2SSCH2CH2NH2), bonded at one terminal amino group to C$^6$. This step converts the inosine to adenine. This chemical reaction was done as described[16], however, with a change for the steps done under nitrogen gas to block atmospheric oxygen done here under a layer of mineral oil. The 21-mer primer (5′-ACAGTCCCTGTTCGGGCGCCG-3′) was synthesized by Integrated DNA Technologies and annealed with 28-mer template as described[16].

The cross-linking of RT139A to the second-base overhang of this T/P was done on a preparative scale without any primer extension. A slight (1.06:1) molar excess of T/P to RT139A was used. A volume of 1.027 ml of solution was prepared to contain 238.6 nmoles (27.7 mg, 232 μM) RT139A, 253.6 nmoles (247 μM) annealed template-primer, 5 mM MgCl$_2$, 1 mM βME, in a total of 70 mM NaCl and 50 mM Tris–HCl pH 8.0. This mixture was incubated at 37 °C for 4 h, then placed on ice overnight. Unreacted RT was removed by linear NaCl gradient purification using HiTrap heparin cartridge (GE Healthcare). The OD$_{260}$:OD$_{280}$ and OD$_{260}$:OD$_{230}$ ratios were measured for all fractions and found to be ~1.14 and ~0.36 respectively, for cross-linked RT in both wash and eluted peaks. These numbers are ~0.54 and ~0.13 for the free RT fraction. The two cross-linked peaks (wash and eluted) were pooled, concentrated, and buffer-exchanged to 10 mg/ml in 10 mM Tris-HCl pH 8.0, 75 mM NaCl.

The cross-linked RT139A/DNA complex was set up in random crystallization screening trials. The 96 conditions of the BCS screen (Molecular Dimensions) at 50% concentration were set up at 4 °C in an MRC-SD2 tray with each well containing 50 μl of precipitant. About 0.1 μl of the protein/DNA complex at 10 mg/ml in 10 mM Tris-HCl, 75 mM NaCl was mixed with an equal volume of the well solution in a sitting drop setup. High-throughput crystallization and crystal screening were done using a Mosquito liquid dispensing robot, SPT Labtech Inc., and a JENSi UVEX microscope attached with a robotic arm. Condition H10 (diluted to 50%) gave chunky crystals; the original H10 condition had 0.2 M (NH$_4$)$_2$SO$_4$, 10 mM cadmium chloride, 0.1 M PIPES, pH 7.0, 15% v/v PEG Smear Broad, and 10% v/v ethylene glycol. The H10 crystallization hit was optimized to the precipitant solution containing 11–12% v/v PEG Smear Broad, 10% w/v sucrose, 50 mM PIPES-NaOH pH 6.5, 0.1 M (NH$_4$)$_2$SO$_4$, 5 mM MgCl$_2$, and 5 mM CdCl$_2$ that produced crystals typically measuring 150 × 45 × 30 μm in size and the crystals were tested for their X-ray diffraction quality and the diffraction data for the apo structure were collected from one such crystal. For fragment screening, a large number of crystallization drops (0.1 + 0.1 μl) were setup on MRC-SD2 trays using the mosquito robot. The crystals were grown over 85 μl precipitant in a well at 20 °C. The crystals were transported to the XChem facility for fragment-screening experiment.

**RT expression and RT/37-mer DNA-aptamer complex preparation for cryo-EM**. For cryo-EM studies, we used an RT construct that has D498N-mutation (RT127A) at the RNase H active site as previously reported[38]; the D498N mutation blocks RNase H activity, but exhibits polymerase activity comparable to wild-type RT. The expression and purification of D498N mutant RT was carried out following a previously described protocol[38]. Briefly, E. coli BL21-CodonPlus (DE3)-RIL cells harboring RT expression construct were allowed to grow until OD$_{600}$ reached 0.9, induced with 1 mM IPTG, followed by further shaking for 3 h at 37 °C. The cells were harvested, lysed by sonication (Branson Sonifier SFX250), and clarified by centrifugation. The cleared lysate was loaded onto a 5 ml Ni-NTA column (GE Healthcare) connected to a FPLC system (GE Healthcare) and purification was performed following the

manufacturer's instructions. A final ion-exchange purification using a Mono-Q column (GE Healthcare) was carried out, and the protein was buffered exchanged into 10 mM Tris-HCl pH 8.0 and 75 mM NaCl.

The 37-mer hairpin-DNA aptamer (5′-TAATAT<u>C</u>CCCCCTTCGGTGCTTT GCACCGAAGGGGGG-3′) for the complex with the R<u>T</u> was ordered from Integrated DNA Technologies; two 2′-O-methylated Cs are underlined. The aptamer pellet was resuspended in 10 mM Tris pH 8.0, 50 mM NaCl, and 1 mM EDTA, and added to RT in 1:1.2 protein-to-aptamer molar ratio. The mix was allowed to incubate for 1 h over ice and loaded to a Superdex 200 Increase 10/300 GL size exclusion column (GE Healthcare) preequilibrated in buffer containing 10 mM Tris-HCl pH 8.0, 75 mM NaCl. The major peak eluted at retention volume of ~13 mL contained the RT-aptamer complex as confirmed by $OD_{260}:OD_{280}$ measurement and peak that shift relative to unbound RT (Supplementary Fig. 5). The complex was aliquoted and stored at −80 °C until further use.

**Fragment screening, X-ray data collection, and structure solution.** Screening of fragments on cross-linked RT crystals was performed at one-of-a-kind high throughput facility Xchem[39] at Diamond Light Source, UK. The semiautomated platform and experimental steps involved in this X-ray crystallography-based fragment screening schematically summarized in Supplementary Fig 2. Briefly, 300 fragments were transferred into 300 crystal drops, and the crystal soaking was done for 1–3 h at 20 °C in fragments at concentrations between 10 and 30 mM. The fragment was selected from DSi-Poised[18] and Fraglites library[19] that was stored at 100 mM concentration in ethylene glycol, and dispensed into the crystal drops using an acoustic dispenser[40]. Crystals were mounted manually on Litholoops (Molecular Dimensions, Sheffield, UK) and diffraction datasets were collected at beamline I04-1 in unattended mode. Data sets were processed using automated processing pipelines available at Diamond Light Source. After inspection, thirty best datasets were processed manually using iMosflm[41], and structures were solved using Phaser in the PHENIX package[42], then peaks in 2Fo–Fc and Fo–Fc maps were inspected for fragment density.

**Docking methodology.** Chemical structures of designed ligands were drawn using ChemBioDraw Ultra 14.0, energy minimized by MM2 method, and saved in pdb format. Asymmetric unit copy of the RT/dsDNA exhibiting P-pocket was selected out from the cocrystal structure with **166**. The receptor grid was defined around the fragment-binding site, as dictated by the **166** cocrystal structure, using Auto-Dock Tools 1.5.4 graphical interface[43]. The grid-box size was set to 22 × 22 × 18 xyz points and its center designated at dimensions (x, y, and z) −129.599, −2.922, and 13.379. After imparting partial atomic charges, the receptor was saved in pdbqt file format. Ligands were also imparted with partial charges and saved in pdbqt format. Docking of the ligands was performed with AutoDock Vina ver. 1.1.1[44] with default settings and the obtained poses were analyzed in PyMol[45] to assess protein–ligand interactions (Supplementary Figs. 3 and 4). The conformation with the most favorable free energy of binding was selected (Supplementary Table 3). 2-D plots of interactions were obtained using LigPlot[46].

**Cryo-EM sample preparation.** RT/DNA-aptamer complex with fragments **166** and **F04** was freshly prepared for cryo-EM application. About 100 mM stock solutions of fragments **166** or **F04** in 100% ethylene glycol were diluted down to 10 mM working stocks in a buffer containing 10 mM Tris-HCl pH 8.0 and 75 mM NaCl. About 370 μM of the fragment was added to 0.4 mg/ml (3.1 μM) RT-aptamer complex in 10 mM Tris pH 8.0, and 75 mM NaCl (1:120 protein to fragment molar ratio) and allowed to incubate for 2 h on ice for complex formation. Post incubation, the samples were immediately used for cryo-EM grid preparation.

**Cryo-EM grid preparation, data collection, and processing.** Vitreous cryo-EM grids for RT/DNA aptamer/**166** and RT/DNA aptamer/**F04** complexes were prepared on Quantifoil R 1.2/1.3 holey-carbon grids. The grids were precleaned with chloroform for 2–3 h, dried overnight, and glow discharged for 1 min at 10 mA with the chamber pressure set at 0.30 mBar (PELCO easiGlow, Ted Pella). The grids were mounted in the sample chamber of a Leica EM GP set at 8 °C and 95% relative humidity. The optimized grids were obtained by spotting 3 μl of the sample at 0.4 mg/ml, incubating for 30 sec, back-blotting for 14 sec using two pieces of Whatman Grade 1 filter paper, and plunge-freezing in liquid ethane at temperature −172 °C. The grids were clipped and mounted on a 200 kV Glacios TEM with autoloader and Falcon 3 direct electron detector as installed in our laboratory.

High-resolution data sets for both fragment complexes were collected on the Glacios using EPU software version 2.9.0 (ThermoFisher Scientific). Electron movies were collected in the counting mode at a nominal magnification of 150,000x yielding a pixel size of 0.97 Å. The total exposure time was 55 s with a total dose of 50 e⁻/Å² in 40 frames and the movies were recorded as gain corrected MRC files. The data-collection parameters are listed in Supplementary Table 4.

**Cryo-EM data processing and model building.** All frames in individual movies were aligned using MotionCor2[47] as implemented in Relion-3.1 and contrast-transfer function (CTF) estimations were performed using CTFFIND-4[48]. The particles were picked using the Laplacian-of-Gaussian autopicking routine in Relion-3.1. Good particles were selected using 2D-class averaging and 3D classification. Initial low-resolution reference map for the 3D classification was generated from the crystal structure of RT/aptamer DNA (PDB Id. 5HP1) using Chimera[49]. The final set of particles for each structure was generated after cycles of 2D and 3D classifications, and the particles were re-extracted. The gold-standard autorefined maps were further improved by B-polishing and CTF refinement. The final autorefined map for each structure was B-sharpened using the post-processing routine in Relion-3.1. All data processing was carried out by Relion-3.1. The final maps were obtained at 3.38 Å and 3.58 Å, respectively, for RT/aptamer DNA/**166** and RT/aptamer DNA/**F04** structures (Supplementary Fig. 9).

The RT/aptamer DNA (PDB Id. 5HP1) crystal structure was used to build the atomic model into the 3.38 Å resolution RT/aptamer DNA/**166** complex. The model building was done manually using COOT[50]. The real-space refinement of the model to the density map was carried out using Phenix 1.19[51]. The structure figures were generated using PyMOL (https://pymol.org/2/) and Chimera[49]. The atomic model for RT/DNA aptamer/**F04** complex was obtained by fitting the RT/aptamer DNA/**166** complex structure to the 3.58 Å density map. The RT/DNA aptamer/**F04** structure was refined by following the steps used for the RT/aptamer DNA/**166** structure. The fragment models for the respective complexes were built into the experimental density in P-pocket and the atomic B factors for individual fragments are comparable to that of the surrounding residues in the respective structures.

**In vitro RT-inhibition assay.** The RT inhibition assay was carried out using a Cy5-flurophore-labeled 17-mer primer (5′- Cy5-CAGGGAAACAGCTATGAC) and a template (5′-TTTTTTTGTCATAGCTGTTTCCTG-3′). The enzymatic reaction mixture contained 125 nM primer-template complex in 50 mM Tris−HCl pH 8.3, 3 mM MgCl2, 10 mM DTT, and 5 μM dATP. Efavirenz (positive control) or RMC compounds, and 0.06 μg of RT per 1 μL reaction (0.2 μl of RT139A at 6.1 mg/ml concentration was used in a 20 μl reaction). The primer and template were preannealed at 1:2 molar ratio by heating up at 95 °C and cooling down to room temperature. The reaction mixture without dATP was preincubated at 37 °C for 20 min. The reaction was initiated by adding the dATP and quenched after 10 min by adding a double volume of quenching buffer (90% formamide, 50 mM EDTA, and 0.05% orange G) and heated at 95 °C for 5 min. The samples were separated on a 1 mm 15% denaturing poly-acrylamide gel, and gel bands were visualized using the Typhoon FLA 9500 imaging system (GE Healthcare). The images were processed using ImageQuant TL v8.1.0.0 (GE Healthcare). The inhibition data are shown as Fig. 5g. A control gel with ladder and the raw gel picture are shown in Supplementary Fig. 10.

**Reporting summary.** Further information on research design is available in the Nature Research Reporting Summary linked to this article.

## Data availability

The data that support this study are available from the corresponding author upon reasonable request. The coordinates and structure factors for the crystal structures of I63C RT/DNA, and its complexes with the fragments 048 and 166, are deposited in Protein Data Bank (PDB) with accession codes 7OZ2, 7OXQ, and 7OZ5, respectively. The coordinates and cryo-EM density maps for the structures RT/DNA aptamer/**166** and RT/DNA aptamer/**F04** complexes are deposited with PDB accession codes/EMDB codes 7OZW/EMD-13139 and 7P15/EMD-13156, respectively.

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

## Acknowledgements

We are grateful to the staff of the XChem facility and I04-1 beam line at Diamond Light Source, UK, Eddy Arnold for plasmids, and Brent De Wijngaert for help with the cryo-EM data collection. W.G. acknowledges the China Scholarship Council (CSC) for funding (grant no. 201707060007). The study was supported by Rega Virology and Chemotherapy internal grants to K.D. The fragment-screening work at XChem has been supported by iNEXT, grant number PID5281, funded by the Horizon 2020 program of the European Union.

## Author contributions

Experiment design and execution, A.S., S.E.M., W.G., H.N., D.S., P.H., S.D.J, and K.D.; data curation, A.S., S.E.M., S.D.J., and K.D.; supervision, A.S., S.D.J., and K.D.; writing and editing, A.S., S.E.M., W.G., D.S., P.H., S.D.J., and K.D.; project conceptualization and administration K.D.

## Competing interests

The authors declare no competing interests.

## Additional information

**Peer-review information** *Nature Communications* thanks the anonymous reviewers for their contribution to the peer review of this work. Peer-reviewer reports are available.

