## [Peer Review File · Nature Communications]

Sliding of HIV-1 reverse transcriptase over DNA creates a transient P pocket – Targeting P-pocket by fragment screeningReviewers' Comments:

Reviewer #1:

Remarks to the Author:

Reverse transcriptase is an essential enzyme for the replication of HIV, and the target of multiple therapeutics. RT has been a rich target for structural and biophysical characterization, with numerous high resolution structures of RT bound to DNA-DNA, DNA-RNA and RNA-RNA substrates. Single-molecule and other data on RT-DNA and RNA complexes have highlighted the dynamic nature of RT interaction with their substrates. In their manuscript, Singh, et al. present structural work with reverse transcriptase complexes crosslinked to a DNA double helix primer-template complex to capture a "P-pocket state." In this complex, the primer 3' end is shifted to the P-1 position, leaving an empty p-site, where the primer terminus usually resides. In the crystal structure, two states in the asymmetric unit—the P-1 state and a 2nd RT-DNA complex with the primer at the N state, which represents a pretranslocation state. Thus, both states are not in an active form for RT-catalyzed dNTP incorporation. The co-existence of the two states can be attributed to the cadmium ions that have replaced magnesium cations in the active site, rendering the N complex inactive, and sliding of the duplex within the RNaseH domain in one of the two complexes. The structural studies are well performed and interesting, although the presence of Cd²⁺, a strong inhibitor, dampens the biological relevance.

P-1 complex crystals were then used for fragment screening to find molecules that would bind to the P-pocket state. Two crystallographic datasets emerged with clear density for each fragment, named 048 and 166, in the P-pocket. 048 and 166 occupied the P-site in slightly different areas and had aromatic rings that allowed for pi-pi stacking interactions. Fragment 048 binding resulted in minimal active site rearrangements and some local shifts in the DNA duplex. Conversely, fragment 166 bound to a slightly different part of the P-pocket, resulting in pocket expansion and active site rearrangements. One of the interactions 166 was making is reminiscent of a base pair, even though its incapable of making one, which inspired second generation fragment design to increase binding affinity. A few fragments emerged that could make base pair-like interactions with the template DNA. To show how they bound to the P-1 complex, additional P-1 complex structures were solved using cryoEM with fragment 166 and fragment F04, one of the second-generation fragments. The structures showed that F04's increased binding affinity is likely higher due to its more planar geometry as well as its capability to form both pseudo base pairing and increased stacking interactions that increased binding affinity. Even with these increased interactions, reverse transcription assays demonstrated that the fragments are weak inhibitors of RT but provide evidence that these P-pocket interactions can be targets for RT inhibition in the future.

While this work has potential interest to readers of Nature Communications, it is a bit narrow. I would have liked to see stronger RT inhibition. That said, this is a nice combination of RT structural biology, and drug screening both real and virtual. The need for polymerase inhibitors that capture transient states is deep and exciting. To consider for publication, the authors should address the following comments.

In line 78, please consider specifying that crosslinked complex cocrystals were used. That would help let readers know what kind of data they are about to see in the results section and provide evidence for the transient nature of the P-pocket state.

1. In figure 1 panels b-d, what are the distances for the crosslink in the structures? It could just be the POV used to generate the figures but the bond in (d) looks a bit large. Panel (e) would be a nice place to do so.
2. The figure callout in figure 156-159 is a bit confusing. I think sticking to the SI figure 1 callout would highlight the intercalating interaction at the crystal symmetry interface more clearly.
3. Please elaborate on what kind of electron density is being shown for 048 and 166 in figure 2. Is this an Fo-Fc or a 2Fo-Fc map? Also, what contour is being used in this figure?
4. Binding affinity is mentioned throughout this paper but qualitative binding affinity statements are made, even though the measurements have been made. One way to make the main text stronger is

to make quantitative references the binding affinity measurements in SI table 3. Lines 204-207 and 230-232 are two places where this would strengthen the manuscript.

a. Considering the binding affinities, could you briefly mention the occupancy for the fragments in the crystal structures?

5. Could you elaborate on the rationale for cryoEM experiments? You described the experimental design very well, but I don't completely understand the motivation behind the switch from cocrystallization to cryoEM.

6. For the cryoEM datasets, how many 3D classes did each complex have and how populated were these classes? Were any classes seen where the fragment wasn't bound, or the complex wasn't in the P-1 state? In general, more detail about the final cryoEM work is needed.

7. The authors should please elaborate on the decision exclude fragment 048 in the second round of design?

8. Comparing the crystal and cryoEM structures, did the contact landscape change or were the rearrangements slight enough that the same contacts were present in both types of structures?

9. Panel f of figure 5 is missing a callout in the manuscript. Lines 308-310 seem like be a good place to reference this panel of the figure.

a. Additionally, could you please include the P-pocket RMSD in the figure legend?

10. Considering your functional data, could the authors please briefly describe the decision to choose compound F04 chosen for structural analysis? If any other compounds were also screened, they should describe how that data compares to F04?

11. The authors glaringly omit any discussion of RT initiation complexes, which may be an outstanding target for their inhibitors and approach. RT initiation complex structures by Viani Puglisi and Arnold have shown dynamics in the position of the 3' end that could be harnessed by the author's inhibitors. Discussion of this should be added.

Reviewer #2:

Remarks to the Author:

In this report, Singh et al. identified a unique crystal form of HIV-RT using a previously described construct of RT crosslinked to a dsDNA substrate. The 2 copies of HIV-RT/dsDNA in the asymmetric unit formed crystal packing contacts at the dsDNA duplex interface which led to a unique positioning of HIV-RT on the duplex. One copy contained a "blocked" N-site in which the 3' primer end was residing in the incoming nucleotide binding site. In the other copy, the polymerase was translocated to a position in which the 3' primer end was at the P-1 position, which ultimately revealed a transient pocket with potential for ligand binding. Given this information, Singh et al. conducted a fragment screen to search for drug-like molecules that could interact at this site. Two fragments were identified, and subsequent virtual screening, docking, and synthesis were conducted with goal of improving ligand binding interactions at the site. In parallel, fragments of interest were also investigated by single-particle cryo-EM using an HIV-RT/dsDNA aptamer construct which provided evidence that the transient pocket can form in solution (without crystal packing artifact) and was competent for fragment binding. The authors report that two fragment binders demonstrate HIV-RT inhibition in a polymerase assay, although this data is somewhat weak and only qualitative. Nevertheless, this work is likely to be of significant interest to the field as it provides evidence for a novel transient pocket that could provide differentiated mechanism of action for HIV-RT inhibitors. Furthermore, it highlights the potential for identification of cryptic or transient pockets across other viral polymerases through combined structural and biochemical/biophysical techniques.

A few minor suggestions for the authors to consider:

1) Line 99. A 27/20-mer template/primer dsDNA is labeled in text, but methods state 28/21-mer, and figure 1a is 27/21-mer. Please check for consistency.

2) It could be helpful to include distances in figure 1e between N6 of adenine and I63. This would give

the reader a better sense of how strained the template strand is by the HIV-RT translocation.

3) Polder maps were used to improve what was likely poor quality Fo-Fc ligand difference maps for structures of 166 and 048. Compound 048 contains bromine and the X-ray data was collected at bromine remote wavelength. The authors could consider calculating the anomalous difference map for 048 and overlay with the polder map to provide additional confidence in the ligand fit.

4) The ligand density from cryo-EM structures is very difficult to evaluate in figure 5a and 5d. A separate figure with ligand and density cutout would be helpful as a supplemental figure.

We highly appreciate the reviewers for spending valuable time in going over the manuscript in very detail and adding valuable comments. We are responding below each comment.

Reviewer #1 (Remarks to the Author):

Reverse transcriptase is an essential enzyme for the replication of HIV, and the target of multiple therapeutics. RT has been a rich target for structural and biophysical characterization, with numerous high resolution structures of RT bound to DNA-DNA, DNA-RNA and RNA-RNA substrates. Single-molecule and other data on RT-DNA and RNA complexes have highlighted the dynamic nature of RT interaction with their substrates. In their manuscript, Singh, et al. present structural work with reverse transcriptase complexes crosslinked to a DNA double helix primer-template complex to capture a “P-pocket state.” In this complex, the primer 3’ end is shifted to the P-1 position, leaving an empty p-site, where the primer terminus usually resides. In the crystal structure, two states in the asymmetric unit—the P-1 state and a 2nd RT-DNA complex with the primer at the N state, which represents a pretranslocation state. Thus, both states are not in an active form for RT-catalyzed dNTP incorporation. The co-existence of the two states can be attributed to the cadmium ions that have replaced magnesium cations in the active site, rendering the N complex inactive, and sliding of the duplex within the RNaseH domain in one of the two complexes. The structural studies are well performed and interesting, although the presence of Cd²⁺, a strong inhibitor, dampens the biological relevance.

P-1 complex crystals were then used for fragment screening to find molecules that would bind to the P-pocket state. Two crystallographic datasets emerged with clear density for each fragment, named 048 and 166, in the P-pocket. 048 and 166 occupied the P-site in slightly different areas and had aromatic rings that allowed for pi-pi stacking interactions. Fragment 048 binding resulted in minimal active site rearrangements and some local shifts in the DNA duplex. Conversely, fragment 166 bound to a slightly different part of the P-pocket, resulting in pocket expansion and active site rearrangements. One of the interactions 166 was making is reminiscent of a base pair, even though its incapable of making one, which inspired second generation fragment design to increase binding affinity. A few fragments emerged that could make base pair-like interactions with the template DNA. To show how they bound to the P-1 complex, additional P-1 complex structures were solved using cryoEM with fragment 166 and fragment F04, one of the second-generation fragments. The structures showed that F04’s increased binding affinity is likely higher due to its more planar geometry as well as its capability to form both pseudo base pairing and increased stacking interactions that increased binding affinity. Even with these increased interactions, reverse transcription assays demonstrated that the fragments are weak inhibitors of RT but provide evidence that these P-pocket interactions can be targets for RT inhibition in the future.

While this work has potential interest to readers of Nature Communications, it is a bit narrow. I would have liked to see stronger RT inhibition. That said, this is a nice combination of RT structural biology, and drug screening both real and virtual. The need for polymerase inhibitors that capture transient states is deep and exciting. To consider for publication, the authors should address the following comments.

In line 78, please consider specifying that crosslinked complex cocrystals were used. That would help let readers know what kind of data they are about to see in the results section and provide evidence for the transient nature of the P-pocket state.

We have now modified the sentence in page 4 as “In this study, we crystallized HIV-1 RT/dsDNA cross-linked complex¹⁶ in a new crystal form with two copies of the complex present in the crystallographic asymmetric unit”

1. In figure 1 panels b-d, what are the distances for the crosslink in the structures? It could just be the POV used to generate the figures but the bond in (d) looks a bit large. Panel (e) would be a nice place to do so.

The N6 – Sγ (Cys63) linker distances in P, N, and P-1 complexes are 5, 4.2, and 5.7 Å, respectively; the distance in P-1 complex is longest among three complexes yet, 5.7 Å can be spanned by a N – C – C – S – S linker.

This information is now also added to the section “I63C cross-link permits sliding of RT over a dsDNA substrate” in page 7. The cross-link distances are now shown in Fig. 1a-e.

2. The figure callout in figure 156-159 is a bit confusing. I think sticking to the SI figure 1 callout would highlight the intercalating interaction at the crystal symmetry interface more clearly.

Sorry for the confusion. We have now removed the reference to Fig. 1a and modified the sentence in page 8 as:

“The 3'-end of the template has a dGMP overhang beyond the DNA duplex region, and the guanine base of this nucleotide in the first copy is intercalated between the duplex DNA and the 3'-end dGMP of the second copy”

3. Please elaborate on what kind of electron density is being shown for 048 and 166 in figure 2. Is this an Fo-Fc or a 2Fo-Fc map? Also, what contour is being used in this figure?

The Fig. 2 map now shows the Fo – Fc difference maps for both fragments. The maps are contoured at 2.5σ for both fragments, **048** and **166**. The polder map density for the fragments **048** and **166** in Fig. 3b and 4b are contoured at 6 and 4.5σ , respectively. The figure legends are now updated with sigma values.

The polder map calculation excludes the effect of bulk solvent correction around the omitted region (the ligand binding area) which in general, improves the electron density compared to an Fo – Fc map; traditionally, the Fo – Fc map is calculated with bulk solvent correction. We also see an improved quality of polder map over the traditional difference map.

4. Binding affinity is mentioned throughout this paper but qualitative binding affinity statements are made, even though the measurements have been made. One way to make the main text stronger is to make quantitative references the binding affinity measurements in SI table 3. Lines 204-207 and 230-232 are two places where this would strengthen the manuscript.

We appreciate the reviewer picking up this point. Since the binding affinity values shown in SI Table 3 for various ligands were derived from in-silico docking studies and were not experimentally determined, we refrained ourselves from making a quantitative statements. We have rather used the values to make relative comparison among the designed fragments to assess their improvement over the parent fragment **166**. To avoid any confusion, we have substituted “..improve the binding affinity” with “improve binding”.

a. Considering the binding affinities, could you briefly mention the occupancy for the fragments in the crystal structures?

The B-factors of the fragments and surrounding residues are highly comparable; the average B-factor of the fragment **048** is 79.60 Å² while the P-pocket residues have an average B-factor of 75.67 Å², and the average B-factor of the fragment **166** is 122.30 Å² while the pocket has an average B-factor of 108.39 Å². This data suggests that the P-pocket is flexible yet, nearly fully occupied by the fragments.

This information is now added in page 11.

5. Could you elaborate on the rationale for cryoEM experiments? You described the experimental design very well, but I don't completely understand the motivation behind the switch from cocrystallization to cryoEM.

We have now modified the first paragraph in the section “Cryo-EM structures of RT/DNA aptamer with **166** and **F04**” in page 13 as:

In our fragment screening experiment, we could get only thirty X-ray diffraction datasets from 300 fragment-soaked crystals (Fig. 2). In our subsequent studies, we experienced that often the crystals are either dissolved or losing diffraction quality in soaking experiments that hinder our ability to routinely and reliably obtaining the structures for this drug-design project. It is likely that soaking of the compounds interfere with the Cd²⁺ interactions and destabilize the crystals. To overcome this experimental limitation, we investigated the possibility of using single-particle cryo-EM on the project. The cryo-EM structures are now free from the impact of Cd²⁺ and crystal contacts.

6. For the cryoEM datasets, how many 3D classes did each complex have and how populated were these classes? Were any classes seen where the fragment wasn't bound, or the complex wasn't in the P-1 state? In general, more detail about the final cryoEM work is needed.

We highly appreciate this question. Now we have added 3D classes for the RT/DNA/**166** complex to Supplementary Fig. 9. The 3D classifications were done in two steps. We were watchful for possible existence of multiple structural states in our complex. However, our 3D classifications in current structures only helped extracting a single set of homogenous particles and no other stable conformation of the complex was observed.

7. The authors should please elaborate on the decision exclude fragment 048 in the second round of design?

The structures of two fragments bound at two parts of the pocket indicated multiple optimization opportunities that we want to explore systematically. We started our post-fragment-soaking study by optimizing the fragment **166** because (i) it occupies a larger part of the pocket, (ii) its binding caused a larger structural rearrangement of the P-pocket than **048**, and (iii) **166** interacts with highly conserved structural elements, YMDD motif and primer grip. Following the optimization of 166 backbone, we plan to expand toward the 048 binding site.

This discussion is now added to the section “Fragment-based design - in-silico docking and synthesis” in page 12.

8. Comparing the crystal and cryoEM structures, did the contact landscape change or were the rearrangements slight enough that the same contacts were present in both types of structures?

We determined the structure of the fragment **166** by both X-ray and cryo-EM to assess the impacts of two distinct structural biology techniques on the binding mode of the fragment. Comparison of two structures show almost similar mode of binding and interactions (Fig. 5c). The phenyl ring is stacked against the DNA duplex, and the interactions of **166** with the YMDD hairpin and with the primer grip are conserved in both structures.

9. Panel f of figure 5 is missing a callout in the manuscript. Lines 308-310 seem like be a good place to reference this panel of the figure.

Thank you for pointing out this omission. We are now referring the Fig. 5f in page 16.

a. Additionally, could you please include the P-pocket RMSD in the figure legend?

The rmsd for the pocket residues is 1.4 Å when 166-bound and apo structures are compared. The rmsd for the pocket residues is 1.2 Å when 048-bound and apo structures are compared. These values are now listed in legends of Fig. 3 and Fig 4.

10. Considering your functional data, could the authors please briefly describe the decision to choose compound F04 chosen for structural analysis? If any other compounds were also screened, they should describe how that data compares to F04?

The compounds F03 – F05 showed almost similar docking scores and all three shared a pyridine ring that replaced the phenyl ring of 166. Only modifications were on the thiazole ring, and F04 was selected as the representative compound for structural study just based on docking score while the inhibition assay was being optimized. We hope the structural knowledge from the binding mode of F04 and the inhibition data for all compounds will help design compounds for the next round of synthesis.

11. The authors glaringly omit any discussion of RT initiation complexes, which may be an outstanding target for their inhibitors and approach. RT initiation complex structures by Viani Puglisi and Arnold have shown dynamics in the position of the 3' end that could be harnessed by the author's inhibitors. Discussion of this should be added.

We have now added a new Supplementary Fig. 8, and the following sentences to the section “Cryo-EM structures of RT/DNA aptamer in complexes with 166 and F04”.

The discussed structures primarily represent states of elongation complex. Structures of HIV-1 RT transcription initiation complexes that bind a dsRNA substrate show significantly different track for dsRNA and the position of primer 3'-end when compared with RT/dsDNA elongation complex.^{34,35,36} A comparison of P-1 state with RT/dsRNA complexes (Supplementary Fig. 8) shows that the primer P-1 nucleotide of dsRNA is positioned ~5 Å away from the DNA primer 3'-end at P-1 position. Due to this structural difference, a potential P-pocket in RT/dsRNA complex will be different from the P-pocket observed in the study. (In page 16)

High-resolution cryo-EM structures of RT/dsRNA representing a minimum transcription initiation complex has been reported recently.³⁴ The structures were achieved by acquiring and processing a significantly large quantity of data. In contrast, our cryo-EM condition optimization was focused on fast data collection and processing which is essential for a structure-based drug-design project. (In page 14)

Reviewer #2 (Remarks to the Author):

In this report, Singh et al. identified a unique crystal form of HIV-RT using a previously described construct of RT crosslinked to a dsDNA substrate. The 2 copies of HIV-RT/dsDNA in the asymmetric unit formed crystal packing contacts at the dsDNA duplex interface which led to a unique positioning of HIV-RT on the duplex. One copy contained a “blocked” N-site in which the 3' primer end was residing in the incoming nucleotide binding site. In the other copy, the polymerase was translocated to a position in which the 3' primer end was at the P-1 position, which ultimately revealed a transient pocket with potential for ligand binding. Given this information, Singh et al. conducted a fragment screen to search for drug-like molecules that could interact at this site. Two fragments were identified, and subsequent virtual screening, docking, and synthesis were conducted with goal of improving ligand binding interactions at the site. In parallel, fragments of interest were also investigated by single-particle cryo-EM using an HIV-RT/dsDNA aptamer construct which provided evidence that the transient pocket can form in solution (without crystal packing artifact) and was competent for fragment binding. The authors report that two fragment binders demonstrate HIV-RT inhibition in a polymerase assay, although this data is somewhat weak and only qualitative. Nevertheless, this work is likely to be of significant interest to the field as it provides evidence for a novel transient pocket that could provide differentiated mechanism of action for HIV-RT inhibitors. Furthermore, it highlights the potential for identification of cryptic or transient pockets across other viral polymerases through combined structural and biochemical/biophysical techniques.

A few minor suggestions for the authors to consider:

1) Line 99. A 27/20-mer template/primer dsDNA is labeled in text, but methods state 28/21-mer, and figure 1a is 27/21-mer. Please check for consistency.

We are sorry for the confusion. The earlier published P-complex had a 27/21-mer dsDNA (corrected in line 99) and a 28/21-mer template/primer was used for the current study. Now this has been clearly stated at the beginning of the section “N and P-1 complexes coexist in crystal” in page 6.

2) It could be helpful to include distances in figure 1e between N6 of adenine and I63. This would give the reader a better sense of how strained the template strand is by the HIV-RT translocation.

We have now listed the N6 – S (I63) distances in Figs. 1 b-e.

3) Polder maps were used to improve what was likely poor quality Fo-Fc ligand difference maps for structures of 166 and 048. Compound 048 contains bromine and the X-ray data was collected at bromine remote wavelength. The authors could consider calculating the anomalous difference map for 048 and overlay with the polder map to provide additional confidence in the ligand fit.

We have now added Fo -Fc difference map densities in Fig.2 whereas the polder maps are shown in Fig 3b and 4b, as before. The difference map densities for both fragments are reliable, as discussed in the response to reviewer 1, Q3. The data were collected at Br K-edge, however, we did not see a distinct difference anomalous peak for Br of the fragment **048** which may be attributed to high noise and weak anomalous signal.

4) The ligand density from cryo-EM structures is very difficult to evaluate in figure 5a and 5d. A separate figure with ligand and density cutout would be helpful as a supplemental figure.

We have now added the cryo-EM density for **166** and **F04** in the new Supplementary Fig. 7.

Reviewers' Comments:

Reviewer #1:

Remarks to the Author:

The authors have rigorously and completely responded to my prior critiques, and the manuscript is therefore acceptable for publication in Nature Communications.

Reviewer #2:

Remarks to the Author:

The authors have addressed all suggestions and concerns raised by reviewers and I recommend the revised version for publication.